# A semi-supervised segmentation network based on noisy student learning for MICCAI FLARE22 Challenge

Nuo Tong[1], Kunru Wang[1], Rui Wu[1], Huiying Yang[1], Hu Huang[1]
[1] Xidian University, Xi'an, Shaanxi 710071, China
kunruwang @stu.xidian.edu.cn

**Abstract.** Abdominal organ segmentation is very important for clinical applications. However, manually annotating organs from CT scans is time-consuming and labor-intensive. Therefore, it is hard to get access to a large amount of annotated data. Semi-supervised learning is an effective method to use unlabeled data to reduce data labeling, which has become a research hotspot. In this work, we adopt the noisy-student learning method, firstly train the teacher model on the manually labeled data and generate pseudo-labels for the unlabeled data through the model, and then train the student model on both of the manual and pseudo-labeled data, continuously iteratively update to produce the final result. Since No new U-Net (nnU-Net) is the state-of-the-art medical image segmentation method and designs task-specific pipelines for different tasks, we adopt 3D nnUNet as the segmentation model during the experiments.

**Keywords:** multi-organ segmentation, semi-supervised learning, noisy-student, pseudo-labels

## 1. Introduction

Accurate and robust segmentation of organs or lesions from medical images plays a crucial role in many clinical applications[1]. With the massive increase in labeled data, deep learning has achieved great success in image segmentation. However, for medical images, the acquisition of annotated data is often expensive. Therefore, the semi-supervised learning method using a small amount of labeled data and a large amount of unlabeled data has become a research hotspot.

In this work, we propose a noisy-student learning method[2], which firstly train the teacher model on the manually labeled data and generates pseudo-labels for the unlabeled data through the model, and then train the student model on the manual and pseudo-labeled data. Iterative update are performed to produce the final result. nnUNet[3] proposes a task-specific medical image segmentation pipeline design framework that automatically configures itself and achieves state-of-the-art results on various medical image segmentation benchmarks. Therefore, both the teacher model and the student model in this work use 3D nnUNet[4].

## 2. Method

### 2.1 Noisy-student learning method

Figure 1 shows the architecture of the proposed model. In this work, the proposed method mainly includes the following steps: 1) train the teacher model by hand-labeled data; 2) generate pseudo-labels for unlabeled data by the teacher model; 3) train students on the hand-labeled and pseudo-labeled data Model.

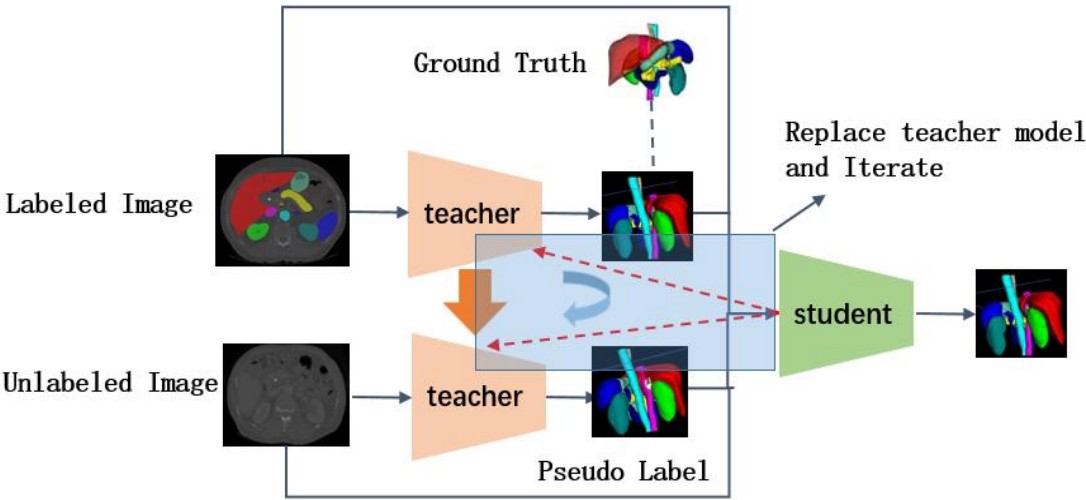

Figure 1: The architecture of the proposed model

## 2.2 nnU-Net architecture

The architecture of the network is shown in Figure 2.

• Network architecture details: The detailed architecture is shown in figure 2. More concretely, we adopt the FabinsUNet as our backbone network, which has been proved to have best performance among the series network of nnU-Net[3].

• Loss function: We use the summation between Dice loss and cross entropyloss because compound loss functions have been roved to be obust n variousmedical image segmentation tasks[5].

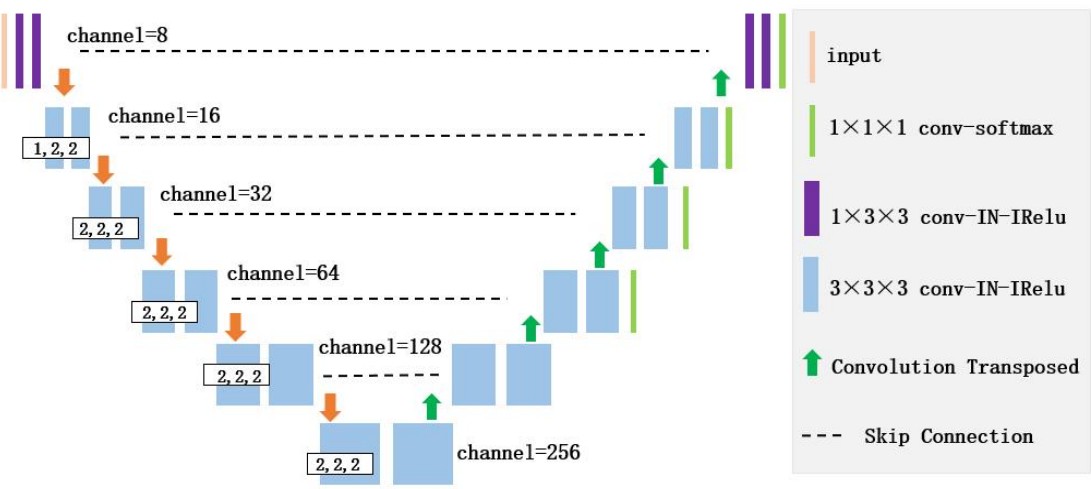

Figure 2: The overall architecture of the nnUNet

## 2.3 Preprocessing

We adopt the same preprocessing procudure as nnU-Net baseline. The following preprocessing steps are conducted:

• Resample:  There are different spacing data in the dataset, unify the spacing of all datasets to (1, 1, 3)

• Crop:  Crop in the non-zero area to reduce computational consumption

• Intensity normalization method:  First, the dataset is clipped to the [0.5, 99.5] percentiles of the

intensity values of the training dataset. Then a zscore normalization is applied based on the mean and standard deviation of the intensity values.

## 2.4 Post-Processing

The whole procedure of post-processing is the same as nnU-net. A connected component analysis of all ground truth labels is applied on training data.

## 3. Dataset and Evaluation Metrics

### 3.1 Dataset

The FLARE2022 dataset is curated from more than 20 medical groups under the license permission, including MSD [10], KiTS [7,8], AbdomenCT-1K [9], and TCIA [6]. The training set includes 50 labelled CT scans with pancreas disease and 2000 unlabelled CT scans with liver, kidney, spleen, or pancreas diseases. The validation set includes 50 CT scans with liver, kidney, spleen, or pancreas diseases. The testing set includes 200 CT scans where 100 cases has liver, kidney, spleen, or pancreas diseases and the other 100 cases has uterine corpus endome-trial, urothelial bladder, stomach, sarcomas, or ovarian diseases. All the CT scansonly have image information and the center information is not available.

The evaluation measures consist of two accuracy measures: Dice Similarity Coefficient (DSC) and Normalized Surface Dice (NSD), and three running effi-ciency measures: running time, area under GPU memory-time curve, and area under CPU utilization-time curve. All measures will be used to compute the ranking. Moreover, the GPU memory consumption has a 2 GB tolerance.

Table 1:Environments and requirements

| Windows version | Ubuntu 18.04.6 LTS |
|---|---|
| CPU | Intel(R) Core(TM) i9-10920X CPU @ 3.50GHz |
| RAM | 64.0GB |
| GPU | One NVIDIA GeForce RTX 3090  24G |
| CUDA version | 11.4 |
| Programming language | Python 3.7 |
| Deep learning framework | Pytorch1.7.0 |
| Specification of dependencies | nnUNet |

Table 2:Training protocols

| Network initialization | "he" normal initialization |
|---|---|
| Batch size | 2 |
| Patch size | $32 \times 224 \times 224$ |
| Total epochs | 1000 |
| Optimizer | Stochastic gradient descent with nesterov momentum ($\mu = 0.99$) |
| Initial learning rate(lr) | 0.01 |
| Lr decay schedule | poly learning rate policy: $(1 - epoch / 1000)^{0.9}$ |
| Training time | 94.5 hours |
| Number of model parameters | 63.07M |
| Number of flops | 1.25T |

| CO2eq | |
| --- | --- |

## 3.2 Evaluation metrics

- Dice Similarity Coeffificient (DSC)
- Normalized Surface Distance (NSD)
- Running time
- Maximum used GPU memory (when the inference is stable)

# 4. Implementation Details

## 4.1 Environments and requirements

The environments and requirements of the proposed method is shown in Table2, and the CPU can be selected in inference process.

## 4.2 Training protocols

The training protocols of the baseline method is shown in Table 3.

## 4.3 Testing Protocols

- Pre-processing steps of the network inputs: The same strategy is applied as training steps.
- Post-processing steps of the network outputs: removing small connected areas to solve the missegmentation problem. And resize the predicted results back to the original size of the predicted image.

# 5. Result

## 5.1 Quantitative results on validation set

Tables 3 and 4 show the results on the validation set, respectively. By comparing the experiments using only labeled data and adding pseudo-labeled data, the effectiveness of the noisy-student learning method is illustrated.

Table 3. Quantitative results on labeld data (validation set)

| Organ | Liver | RK | Spleen | Pancreas | Aorta | IVC |
| --- | --- | --- | --- | --- | --- | --- |
| DSC | 0.9686 | 0.9204 | 0.9186 | 0.8373 | 0.9477 | 0.8753 |
| RAG | LAG | Gallbladder | Esophagus | Stomach | Duodenum | LK |
| 0.7821 | 0.7175 | 0.7325 | 0.8321 | 0.8596 | 0.7549 | 0.8986 |

Table 4. Quantitative results on labeld and psudeo labeld data( validation set)

| Organ | Liver | RK | Spleen | Pancreas | Aorta | IVC |
| --- | --- | --- | --- | --- | --- | --- |
| DSC | 0.9717 | 0.9177 | 0.9233 | 0.8657 | 0.9514 | 0.8618 |
| RAG | LAG | Gallbladder | Esophagus | Stomach | Duodenum | LK |
| 0.7815 | 0.7404 | 0.7564 | 0.8171 | 0.8979 | 0.7639 | 0.8977 |

## 5.2 Quantitative results on validation set

Figure 3 presents some easy examples. It can be seen that the proposed network shows great performance.

Figure 4 presents some challenging examples. The first and second row of Figure 4 illustrates that our proposed method give poor performance in segmenting the left kidney tissue. Third row of Figure 4 shows that our proposed method misclassifies the whole liver tissue. And the fourth row of Figure 4 shows that our proposed method fail to segment the duodenum tissue.

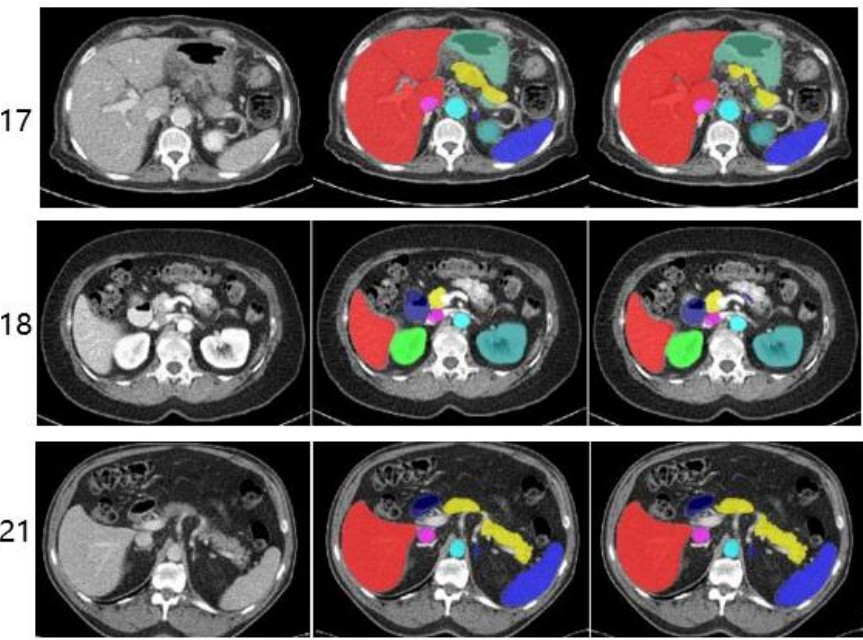

Figure 3: Easy examples. First column is the image, second column is the ground truth, and third column is the predicted results by the proposed method.

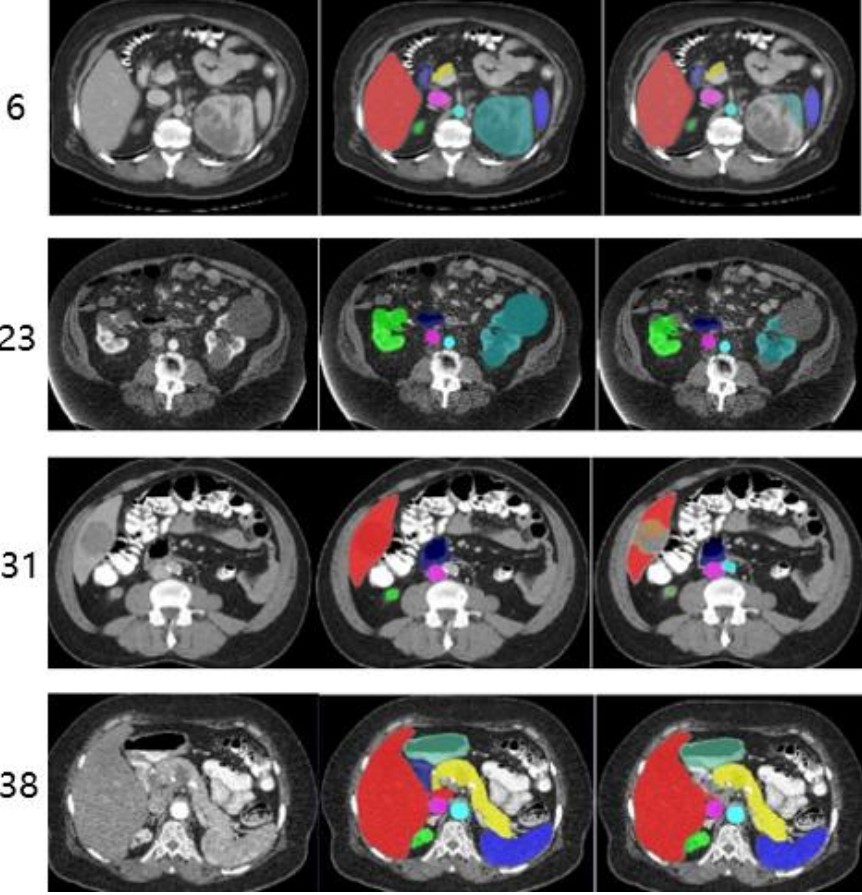

Figure 4: Challenge examples. First column is the image, second column is the ground truth, and third column is the predicted results by the proposed method.

# 6. Discussion and Conclusion

This method can be applied well in cases of healthy or slightly diseased affected organs. This method achieves high generation capacity for segmentation of larger organs such as liver, kidney, and spleen in terms of DSC scoring. Segmentation of soft-tissue organs such as pancreas, duodenum, adrenal gland, etc. performed disappointingly due to anatomical variability in volume and shape between patients. The presence of organs affected by severe lesions is a key factor in poor segmentation performance. Furthermore, further research is required to obtain an accurate boundary segmentation.

## Acknowledgment

The authors of this paper declare that the segmentation method they implemented for participation in the FLARE challenge has not used any pre-trained models nor additional datasets other than those provided by the organizers.

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
