# OpenReview forum: "A semi-supervised segmentation network based on noisy student learning for MICCAI FLARE22 Challenge"
_MICCAI.org/2022/Challenge/FLARE_

### Official Review · Reviewer_rVVg · 2022-09-19
**Well established approach and strong segmentation results, however missing important details.**

**Rating:** 6
**Confidence:** 4

**Review:**

Summary:

This paper combines an iterative noisy student approach with nnU-Net. Here a teacher is trained on the labeled data, which is then used to predict pseudo labels on the unlabeled dataset. Both the labeled and pseudo-labeled dataset are subsequently used to train a student data. This process is iterated to arrive at a performance gain.

Pros:

- Makes use of state-of-the-art tools and shows strong performance.
- Ablations show a clear performance gain when using the unlabeled data.

Problems:

- (minor) Figure 1 shouldn't be called architecture, but rather "schematic overview"or similar.
- Figure 1 could be more clear, esp. the arrows are a bit misleading at first.
- Table 3 makes it hard to compare the ablation, would be better if displayed next to each other.
- The resolution of Figures 3 and 4 are a bit low, which makes it hard to inspect segmentation details.
- Missing details on the training protocol. Is the student always initialized at random again, and how many iterations of teacher-student pseudolabeling are used in the end?
- Missing info on inference requirements in terms of GPU memory, speed, etc. as required by organizer's checklist.

The missing details should definitely be addressed. Note: This is the same submission as ID 15.

---

### Official Review · Reviewer_4gof · 2022-09-19
**A Poorly-Written Paper with Insufficient Experiments**

**Rating:** 5
**Confidence:** 3

**Review:**

This paper uses a teacher network to provide pseudo-labels for training a noisy nnU-Net student. Some comparisons are made to show the effectiveness of the proposed method results.



Pros:


+ This paper adopts a nnU-Net-based teacher model to train a noisy student for better performance, which is a good choice.


+ This paper provides qualitative and quantitative results to show the effectiveness of the proposed method.



Cons:


- Insufficient ablation study to show the effectiveness of each design choice, e.g., how much the iterative teacher-student learning contributes to the performance, and what are the running time and GPU costs.

- This paper can briefly introduce the FLARE 22 Challenge in the Introduction.

- Training protocols presented in Sec. 4.2 seems to be unclear as Tab. 3 does not show the training protocols.

- Tab. 3 & 4 should be improved to make the results easier for understanding. For example, the first column is misleading and the average DSC of all the organs should be listed.

- In Fig. 4, it would be better to discuss a little bit why the proposed method fails in these cases.

- Typos in Sec. 2.2: "roved to be obust n variousmedical"

---

### Official Review · Reviewer_JQmk · 2022-09-20
**Abdominal organ segmentation using  nnUNet and noisy student training scheme.**

**Rating:** 5
**Confidence:** 4

**Review:**

Summary:
This paper achieves efficient abdominal organ segmentation using nnUNet and noisy student training scheme.

Advantage:
1. The main idea of this paper is delivered fluently.
2. Authors made detailed

Disadvantage:
1. The novelty of this paper is trivial. Authors did not make any improvements on the noisy student model.
2. The training details of the teacher model are not clear.  How to evaluate the performance of the teacher model?
3. What is the inference time and computational cost? Since the computational resource is limited, it is important to show the cost.
4. The ablation studies are too weak.

---

### Meta-Review · Program_Chairs · 2022-09-28

**Recommendation:** Major Revision
**Confidence:** 5

**Metareview:**

Reviewers raise many concerns and suggestions. Please address all comments in the revised manuscript.